# Building a functional connectome of the *Drosophila* central complex

Romain Franconville*, Celia Beron[†], Vivek Jayaraman*

Janelia Research Campus, Howard Hughes Medical Institute, Ashburn, United States

**Abstract** The central complex is a highly conserved insect brain region composed of morphologically stereotyped neurons that arborize in distinctively shaped substructures. The region is implicated in a wide range of behaviors and several modeling studies have explored its circuit computations. Most studies have relied on assumptions about connectivity between neurons based on their overlap in light microscopy images. Here, we present an extensive functional connectome of *Drosophila melanogaster*'s central complex at cell-type resolution. Using simultaneous optogenetic stimulation, calcium imaging and pharmacology, we tested the connectivity between 70 presynaptic-to-postsynaptic cell-type pairs. We identified numerous inputs to the central complex, but only a small number of output channels. Additionally, the connectivity of this highly recurrent circuit appears to be sparser than anticipated from light microscopy images. Finally, the connectivity matrix highlights the potentially critical role of a class of bottleneck interneurons. All data are provided for interactive exploration on a website.
DOI: https://doi.org/10.7554/eLife.37017.001

*For correspondence:
franconviller@janelia.hhmi.org
(RF);
vivek@janelia.hhmi.org (VJ)

Present address: [†]Department of Neurobiology, Harvard Medical School, Boston, United States

Competing interests: The authors declare that no competing interests exist.

## Introduction

Positioned in the middle of the insect brain, the central complex provides a unique opportunity to obtain mechanistic insights into the way brains build and use abstract representations (*Turner-Evans and Jayaraman, 2016*). Studies in a variety of insects, including locusts, dung beetles, bees and monarch butterflies, have used intracellular recordings to chart maps of polarized light E-vectors in substructures of the region (*Heinze and Homberg, 2007*; *Heinze and Reppert, 2011a*; *el Jundi et al., 2015*; *Stone et al., 2017*), and extracellular recordings from the cockroach have found sensory and motor correlates throughout the region (*Bender et al., 2010*; *Guo and Ritzmann, 2013*; *Ritzmann, 2012*). More recently, calcium imaging experiments in behaving *Drosophila* have shown that both visual and motor cues can update a fly's internal representation of heading (*Seelig and Jayaraman, 2015*). Independently, neurogenetic studies have used disruptions of the normal physiology of the structure to highlight its involvement in a variety of functions, including motor coordination (*Poeck et al., 2008*), visual memory (*Liu et al., 2006*), sensory-motor adaptation (*Triphan et al., 2010*), and short- and long-term spatial memory (*Neuser et al., 2008*; *Ofstad et al., 2011*). It is likely that these tasks rely on the correct establishment and use of an internal representation of heading (*Giraldo et al., 2018*; *Green et al., 2018*).

The scale of the network—a few thousands of neurons in the fly central complex—and the ease of genetic access to individual cell types in *Drosophila melanogaster*, make this circuit tractable with existing theoretical and experimental methods. Detailed anatomy at the light microscopy level (*Hanesch et al., 1989*; *Wolff et al., 2015*; *Lin et al., 2013*) of a significant fraction of the cell types, along with the availability of tools to genetically target these neurons by type (*Wolff et al., 2015*), have fueled the first mechanistic investigations of how the circuit constructs a stable heading representation (*Kim et al., 2017*), and how this representation updates as the animal turns in darkness (*Turner-Evans et al., 2017*; *Green et al., 2017*). Such results and related findings from other insects

**eLife digest** Some of the most evocative discoveries in neuroscience have been those of internal representations, such as neural activity patterns that represent which direction an animal is facing and its place in its surroundings. Understanding how neurons connect to one another to form 'circuits' is crucial to understanding how these circuits maintain such representations.

Many of the design principles that underlie circuit function in the brains of fruit flies apply to other animals. However, fly brains are easier to study because genetic tools can be used on them to selectively activate and image the activity of specific types of neurons. By activating one type of neuron and imaging the activity of another that may be connected to it, we obtain what is called a functional 'connectome': a map of neural connectivity that identifies different pathways that information can flow along.

A region of the fly brain called the central complex is involved in many important behaviors, including navigation and sleep. Researchers know about the types of neurons in the region and about how the activity of some of them changes during different behaviors. However, obtaining the connectome of the central complex would make it easier to understand how the central complex works.

A technique called optogenetics allows specific types of neurons to be activated one at a time by shining light onto them. By imaging the activity of neurons that might be connected to an optogenetically activated neuron, Franconville et al. have now built an extensive – albeit still incomplete – map of the connections within the central complex of fruit flies.

The map reveals two key bottlenecks in the central complex circuit. Firstly, a neuron type in a substructure called the protocerebral bridge controls a lot of the information flowing through the circuit. Secondly, the circuit appears to have very few true 'output' neuron types – Franconville et al. identified only one. These results suggest that however complicated the computations performed by the central complex circuit might be, the output of the circuit, which likely guides the fly's actions, may be much simpler.

Franconville et al. have compiled the mapping results into an interactive website that makes the neuroscientific data both freely available and easily explorable. As researchers perform more such experiments, the new data can be added to the map. This information can be used to constrain theories and inspire new ideas about how the central complex does what it does.

DOI: https://doi.org/10.7554/eLife.37017.002

have also inspired a number of modeling studies aimed at predicting or reproducing physiologically and behaviorally relevant response patterns (*Kakaria and de Bivort, 2017*; *Givon et al., 2017*; *Chang et al., 2017*; *Cope et al., 2017*; *Su et al., 2017*; *Fiore et al., 2017*; *Kim et al., 2017*; *Stone et al., 2017*; *Turner-Evans et al., 2017*). Many of these models make assumptions about connectivity within the central complex based on the degree of overlap at the light microscopy level between processes that look bouton-like and those that seem spiny, which are suggestive of pre- and post-synaptic specializations, respectively. To go beyond those anatomical approaches, we constructed a connectivity map based on functional data, which includes information about whether connections are effectively excitatory or inhibitory. This map will help dissect the function of the central complex by constraining large-scale models and aiding the formulation and testing of new hypotheses. Given the dozens of central complex cell types (known and yet to be discovered) omitted in our dataset, the diversity of neurotransmitters and receptors they express, the mixture of pre- and post-synaptic specializations in their arbors, and the dense recurrence of the network, we see this map not as a full connectome, but as an initial scaffold that will allow new information to be incorporated as and when it becomes available.

The quest to obtain circuit diagrams dates back to Cajal and Golgi (*Azoulay, 1894*; *Pannese, 1999*), who used sparse labeling techniques to reveal neuron morphology and circuit architectures. Anatomical methods based on marking a discrete subset of neurons and imaging them with light microscopy have recently been revived in the form of techniques relying on stochastic genetic labeling (*Livet et al., 2007*; *Hampel et al., 2011*; *Nern et al., 2015*; *Lee and Luo, 2001*; *Chiang et al., 2011*) and photoactivatable fluorophores (*Patterson and Lippincott-Schwartz, 2002*;

*Ruta et al., 2010*). These methods allow the extraction of the detailed anatomy of individual neurons. But even when used in combination with synaptic markers (*NicolaiNicolai̇ et al., 2010*; *DiAntonio et al., 1993*; *Zhang et al., 2002*; *Fouquet et al., 2009*), such methods do not offer definitive evidence of synaptic connections, as they rely solely on the proximity of putative pre- and post-synaptic compartments. Recently, promising trans-synaptic genetic tagging systems (*Talay et al., 2017*; *Huang et al., 2017*) have been developed to address some of these issues. However, none of these approaches provide any insight into the functional properties of potential connections. Despite such shortcomings, light-level microscopy constitutes a good starting point by constraining the search for possible connections within large populations of neurons —at the simplest level, if putative pre- and post-synaptic compartments do not overlap in light microscopy images, they cannot be in synaptic contact.

Electron microscopy (EM) reconstruction is considered to be the gold standard for connectomics (*White et al., 1986*; *Briggman and Bock, 2012*; *Zheng et al., 2017*; *Schneider-Mizell et al., 2016*). Under ideal conditions, it permits the unambiguous identification of synapses between all neurons in a given volume. As powerful as this capability is, the technique also suffers from a few limitations. Acquiring, processing and analyzing the data is still time-consuming. As a result, connectomes from EM data are typically based on data from a single animal. In addition, EM does not permit the identification of neurotransmitter types at a given synapse, nor does it detect gap-junctions in invertebrate tissue, at least at present (*Zheng et al., 2017*). Finally, it can be challenging to assess the strengths of connections between neurons, because it is not yet clear whether the number of synapses predicts the functional strength of the connection.

Functional methods address some of these drawbacks. Simple measures of activity have been used to assess a form of functional connectivity: regions or neurons whose simultaneously recorded activity is correlated—either spontaneously or during a given task—are deemed connected. This has been used with EEG, fMRI and MEG recordings in humans to establish maps at the brain region level (*Salvador et al., 2005*; *Stam, 2004*) and with multi-electrode recordings in monkeys and rodents (for example, [*Gerhard et al., 2011*]). Functional connectivity has also been inferred from correlations or graded changes in the response properties of neurons recorded in different animals, usually in cases where the neurons have overlapping arbors when examined with light microscopy. This approach has been employed to suggest polarized light processing pathways in the central complex of the locust and monarch butterfly (*Heinze et al., 2009*; *Heinze, 2014*). However, such functional methods are correlative and do not provide a causal basis for the inferred connectivity.

To obtain a causal description of functional connectivity—sometimes termed effective connectivity—it is necessary to either stimulate one node of the network while recording from another one, or record both at sufficiently high resolution as to detect hallmarks of direct connectivity. The most reliable approach of this class is paired patch-clamp recording, which identifies connected pairs and their functional properties with a high level of detail (*Huang et al., 2010*; *Yaksi and Wilson, 2010*; *Fişek and Wilson, 2014*), but can only be performed at low throughput (*Jiang et al., 2015*). In recent years, the development of optogenetics has expanded the toolkit for simultaneous stimulation and recording experiments (*Petreanu et al., 2007*). In *Drosophila*, the ease of use of genetic reagents renders such approaches particularly attractive. Combinations of P2X2 (a mammalian purinergic receptor that can be ectopically expressed in *Drosophila* and activated by ATP application) and the genetically encoded calcium indicator GCaMP (*Yao et al., 2012*), P2X2 and patch-clamp recordings (*Hu et al., 2010*), Channelrhodopsin-2 and patch-clamp (*Gruntman and Turner, 2013*), CsChrimson (a red shifted Channelrhodopsin) and CaMPARI (a calcium activity integrator, see [*Fosque et al., 2015*]) and CsChrimson and GCaMP (*Hampel et al., 2015*; *Zhou et al., 2015*; *Ohyama et al., 2015*) have been used in individual studies to investigate a small number of connections. Methods that rely on the genetic expression of calcium indicators to detect potential post-synaptic responses operate at a lower resolution than paired-recordings since they usually establish connectivity between cell types, as defined by the genetic driver lines used, rather than between individual neurons. These methods cannot definitively distinguish connections that are direct from those that might involve several synapses (but see Results/Discussion) and are limited by the sensitivity of the calcium sensors used. Despite these shortcomings, such methods constitute a good compromise as they provide a causal measure of functional connectivity at a much higher throughput than double patch recordings. It is also worth noting that the advantages and limitations of these techniques complement those of serial EM reconstructions. We chose to apply this combination of

optogenetics and calcium imaging on a large scale by systematically testing genetically defined pairs of central complex cell types in an ex vivo preparation, therefore building a large and extensible map of functional connections in the structure at cell-type resolution.

## Cell types and hypothetical information flow in the central complex

The central complex consists of four main neuropiles — the protocerebral bridge (PB), the ellipsoid body (EB, Central Body Lower in other insects), the fan-shaped body (FB, Central Body Upper in other insects) and the noduli (NO) — and at least three accessory neuropiles (also known as the lateral complex) — the gall (GA), the lateral accessory lobe (LAL) and the bulbs (BU) (*Figure 1A* and [*Wolff et al., 2015*; *Lin et al., 2013*; *Hanesch et al., 1989*]). Throughout this manuscript, we denote output  (resp. input) neurons that link central complex neuropiles to neuropiles outside the central complex. Some of the most striking neural elements of the central complex are the *columnar neurons*, which innervate one of the 18 (in *Drosophila*) glomeruli of the PB, one vertical section of either the FB or EB, and one accessory neuropile — a *column* being constituted by the PB glomeruli and FB/EB section. A total of 12 different columnar cell types have been described, with stereotypical correspondences between the PB glomerulus and the EB/FB section. In addition to these 'principal cells', there are a number of neurons innervating multiple columns of one neuropile. These neurons often innervate subdivisions orthogonal to the columns. Moreover, they sometimes also project to neuropiles outside the central complex. This set of neurons includes the ring neurons, which innervate a ring within the EB and an accessory neuropile, and a collection of inputs and interneurons with processes in the FB and PB. From this light level anatomy and putative synaptic polarity, one can derive a hypothetical picture of information flow through the central complex (*Figure 1Bi*):

- Ring neurons provide input to the EB columnar neurons.
- Recurrent connections between EB columnar neurons form and sustain a ring attractor for heading direction
- Information is transferred from the EB columnar system to the FB columnar system via the PB (interestingly, only one columnar neuron type displays presynaptic terminals in the PB, the E-PGs)
- FB columnar neurons also receive inputs in the FB
- All columnar neurons but the E-PGs also receive inputs in the PB
- Interneurons in the PB and FB further interconnect the columns
- All accessory structures are potential outputs (but see *Stone et al., (2017)* for an example of inputs coming from the LAL to the NO in the sweat bee).

We show that this overall flow of information is generally supported functionally for the parts we have tested so far, but with a few potentially important differences (*Figure 1Bii*): the observed connectivity in the PB is sparse, rendering the function of PB interneurons possibly critical; accessory structures are usually input rather than output areas; and, consequently, output channels of the central complex are scarce.

## Results

### A functional connectivity screen

We picked driver lines for functional connectivity mapping by visually inspecting the Janelia Gal4-driver collection (*Jenett et al., 2012*) for strength of expression in the cell types of interest, and sparseness of the expression pattern in the central complex. The 37 driver lines (for 24 cell types) cover the main columnar neuron types (8 of the 11 types described in *Wolff et al. (2015)*) and PB interneurons (3 out of the five in *Wolff et al. (2015)*), a LAL-FB neuron, three types of ring neurons, a Gall-EB neuron columnar in the EB and neurons innervating accessory structures, namely four types of LAL interneuron and three types of neurons connecting the LAL to the noduli. Drivers are listed in *Table 1* and *Table 2*. Neuron types are schematized in *Figure 2A* and *Figure 1—figure supplement 1*. The dataset includes inputs to the EB system, connections between EB columnar neurons, connections in the PB as well as potential inputs and outputs in the LAL, Gall and noduli. Among the types tested, 43 of the 59 anatomically possible connections could be tested with the reagents available. The connectivity of the multitude of cell types within the FB has not been explored: neither FB interneurons (also known as pontine cells), nor FB input neurons are part of this study.

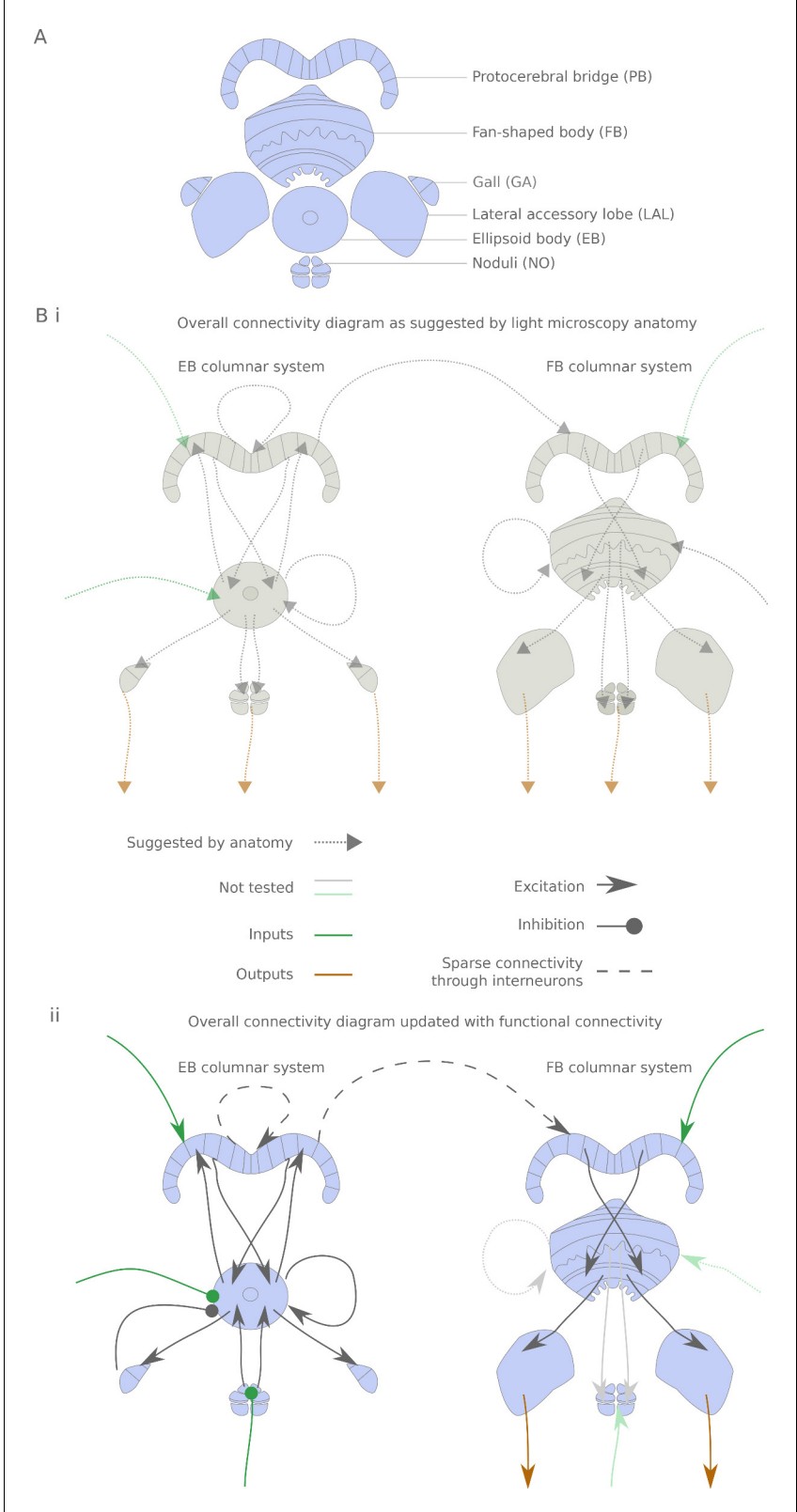

**Figure 1.** The central complex neuropiles and the hypothesized flow of information based on overlap of arbors in light-microscopy images. (A) Schematic representation of the central complex and associated structures used throughout the manuscript. (B) (i) Hypothesized global information flow in the central complex based on neuron morphologies and the overlap of putative pre- and post-synaptic processes between different neuron types,

*Figure 1 continued on next page*

*Figure 1 continued*

based on **Wolff et al. (2015)** and **Hanesch et al. (1989)**. (ii) Connectivity map based on the results of this study. Faded arrows represent hypotheses that were not tested in the present study.

DOI: https://doi.org/10.7554/eLife.37017.003

The following figure supplements are available for figure 1:

**Figure supplement 2.** Same as figure *Figure 1*, but with neuron type names underlying the connections indicated.

DOI: https://doi.org/10.7554/eLife.37017.004

**Figure supplement 1.** Diagrams of all the cell types used in the study.

DOI: https://doi.org/10.7554/eLife.37017.023

Cell-type pairs to be tested were chosen based on overlaps between their expression patterns in light microscopy images. For each combination selected, we expressed CsChrimson and GCaMP6m in potential pre- and post-synaptic partners, respectively (*Figure 2B,C*), and probed their connection in an ex vivo preparation using a standardized protocol (see *Figure 2D*, and Materials and methods). Whenever large responses were observed, we used pharmacology to both check that observed transients were synaptically mediated, and to narrow down the neurotransmitters involved (*Figure 4—figure supplement 2* and *Figure 4—figure supplement 3*).

Effects of stimulation ranged from very large and reliable transients (*Figure 3Ai*) to undetectable changes (*Figure 3Aiii*). In between those extremes, we observed transients of variable size and reliability (*Figure 3B*). To our surprise, we could also detect clear inhibitory responses (*Figure 3Aii*). This was possible because—at least in some cell types—fluctuations in baseline activity occasionally elevated GCaMP levels during the experiment (see Discussion and *Figure 3—figure supplement 3*). Therefore, even though hyperpolarization below resting potential is likely not detectable through calcium imaging, we could detect inhibition from an excited state as a dip in the fluorescence trace.

Since no single characteristic of the responses could adequately capture their variety, variability and complexity, we chose to characterize the transients by using a battery of statistics reflecting response amplitude, shape, reliability and stimulus sensitivity (see *Figure 3C*, *Figure 3—figure supplement 2*, and Materials and methods). Responses of control pairs with non-overlapping processes were then used to form the null-hypothesis distributions of two metrics that capture response amplitude and reliability (see *Figure 3D*). For every data point, the Mahalanobis distance (a covariance corrected measure in a multidimensional space, see Materials and methods) to the null distribution was computed and used as a connection strength metric in summary diagrams like *Figure 4* and *Figure 5*. Non-overlapping pairs usually showed no fluctuations upon stimulation, and when they did, they were small and unreliable (see *Figure 3—figure supplement 1*), likely reflecting effects of indirect connections. Not surprisingly, responses were always detected with same-cell-type-stimulation controls, where CsChrimson and GCaMP6m were expressed in the same neuron type (see *Figure 3D*).

All the individual responses and statistics, in the context of the overall connectivity diagram, are available at http://romainfr.github.io/CX-Functional-Website/romainfr.github.io/CX-Functional-Website/, a website that enables an interactive exploration of the results of this study. We plan to update this website as further experiments are performed. The website can also be expanded to accommodate other sources of data, which would make it an exhaustive source of information about the central complex. The connectivity matrix resulting from our experiments is shown in *Figure 4* in two alternative visualizations, namely a network diagram (*Figure 4*) and a matrix of connection strengths.

## An emerging view of the central complex functional connectome

*Figure 5* outlines some of the connectivity patterns we observed. We focus in particular on inputs and outputs to the ellipsoid body, protocerebral bridge and paired noduli, connectivity within the protocerebral bridge, and components of the ring attractor network within the central complex.

### Inputs

We identified two classes of inhibitory, picrotoxin-sensitive (hence mediated either by GABA-A or Glutamate) inputs to the central complex. First, the two ring neuron types we tested (GB-Eo, L-Ei) target the wedge columnar neurons (E-PG, *Figure 5Ai*), as has been suggested previously (*Martín-*

**Table 1.** Drivers and neuron types used.

When the 'Driver' name is followed by an insertion site, it is a LexA line (see Materials and methods). Names starting by SS are stable splits. All other driver names correspond to both Gal4 (in attP2) and LexA (in attP40) drivers. 'New Type Name' refers to the nomenclature for short names adopted in this paper (following [*Kakaria and de Bivort, 2017*; *Turner-Evans et al., 2017*; *Green et al., 2017*]). Type description is the long name, following the guidelines of *Wolff et al. (2015)*. Pre and post regions are labeled based on anatomical characteristics. The finer subdivisions were used to establish if two neurons were anatomically overlapping. The corresponding name in other insect species were determined using the insect brain database (https://insectbraindb.org/) and related papers (*Heinze and Homberg, 2008*; *Heinze and Reppert, 2011b*; *Stone et al., 2017*)

| Driver | New type name | Type description | Super type | Name in other insects |
|---|---|---|---|---|
| 87G07 | P-F3N2d | PBG2-9.s-FBl3.b-NO2D.b | FB columnar | CPU4 (b or c) |
| 85H06 | P-F1N3 | PBG2-9.s-FBl1.b-NO3PM.b | FB columnar | CPU5 |
| 60D05 | E-PG | PBG1-8.b-EBw.s-DV_GA.b | EB columnar | CL1a |
| SS02191 | P-EG | PBG1-8.s-EBt.b-DV_GA.b | EB columnar | CL1b |
| 67D09 | P-F3N2v | PBG2-9.s-FBl3.b-NO2V.b | FB columnar | CPU4(a) |
| 67D09-attP5 | P-F3N2v | PBG2-9.s-FBl3.b-NO2V.b | FB columnar | CPU4(a) |
| 67D09-VK22 | P-F3N2v | PBG2-9.s-FBl3.b-NO2V.b | FB columnar | CPU4(a) |
| 37F06 | P-EN1 | PBG2-9.s-EBt.b-NO1.b.Type1 | EB columnar | CL2 |
| 37F06-VK22 | P-EN1 | PBG2-9.s-EBt.b-NO1.b.Type1 | EB columnar | CL2 |
| VT008135 | P-EN1 | PBG2-9.s-EBt.b-NO1.b.Type1 | EB columnar | CL2 |
| SS02232 | P-EN2 | PBG2-9.s-EBt.b-NO1.b.Type2 | EB columnar | CL2 |
| 84H05 | PF-LCre | PBG1-7.s-FBl2.s-LAL.b-cre.b | FB columnar | CPU1a |
| 84H05-VK22 | PF-LCre | PBG1-7.s-FBl2.s-LAL.b-cre.b | FB columnar | CPU1a |
| 84H05-attP5 | PF-LCre | PBG1-7.s-FBl2.s-LAL.b-cre.b | FB columnar | CPU1a |
| 55G08 | Δ7 | PB18.s-GxΔ7Gy.b-PB18.s-9i1i8c.b | PB interneuron | TB1 |
| 55G08-attP5 | Δ7 | PB18.s-GxΔ7Gy.b-PB18.s-9i1i8c.b | PB interneuron | TB1 |
| 55G08-VK22 | Δ7 | PB18.s-GxΔ7Gy.b-PB18.s-9i1i8c.b | PB interneuron | TB1 |
| 47G08 | IS-P | PBG2-9.b-IB.s.SPS.s | PB input | |
| 49H05 | IMPL-F | LAL.s-IMP-FBl3.b | FB input | |
| 75H04 | L-Ei | EBIRP I-O-LAL.s | Ring neuron | TL4 |
| 32A11 | L-Em | EBMRP I-O-LAL.s | Ring neuron | TL4 |
| 18A05 | GB-Eo | EBORP O-I-GA-Bulb | Ring neuron | |
| 18A05-VK22 | GB-Eo | EBORP O-I-GA-Bulb | Ring neuron | |
| 17H12 | AMPG-E | EB.w-AMP.d-D_GAsurround | EB input | |
| 12C11 | EFBG | EBMRA-FB-LT-LT-GA-GA | Other | |
| 72H06 | SMPL-L | SMP.s-LAL.s-LAL.b.contra | LAL-IN | |
| 72H06-attP5 | SMPL-L | SMP.s-LAL.s-LAL.b.contra | LAL-IN | |
| 72H06-VK22 | SMPL-L | SMP.s-LAL.s-LAL.b.contra | LAL-IN | |
| SS02615 | SMPL-L2 | SMP.s-LAL.s-LAL.b.contra2 | LAL-IN | |
| 26B07 | WL-L | Wedge-LAL.s-LAL.b.contra | LAL-IN | |
| 31A11 | L-Cre | LAL-Cre | LAL-IN | |
| SS00153 | S-P | SPS.s-PB.b | PB input | |
| 76E11 | GL-N1 | LAL.s-GAi.s-NO1i.b | LAL-NO | |
| 76E11-VK22 | GL-N1 | LAL.s-GAi.s-NO1i.b | LAL-NO | |
| SS04448 | GL-N1 | LAL.s-GAi.s-NO1i.b | LAL-NO | |
| SS04420 | CreL-N2 | Cre.s-LAL.s-NO2.b | LAL-NO | TN1 ? |
| 12G04 | L-N3 | LAL.s-NO3Ai.b | LAL-NO | TN1 ? |

DOI: https://doi.org/10.7554/eLife.37017.005

**Table 2.** Number of cells per hemisphere and PB column (when relevant) in the drivers used in this study, as estimated by counting cell bodies in confocal stacks.
When counting was difficult (e.g. because of densely packed somata) we indicated this by a ~ sign.

| Driver | New type name | Cells per hemisphere | Cells per column |
|---|---|---|---|
| 87G07 | P-F3N2d | 16 | 2 |
| 85H06 | P-F1N3 | 32 | 4 |
| 60D05 | E-PG | 24 | 3 |
| SS02191 | P-EG | 16 | 2 |
| 67D09 | P-F3N2v | 24 | 3 |
| 37F06 | P-EN1 | 24 | 3 |
| VT008135 | P-EN1 | 24 | 3 |
| SS02232 | P-EN2 | 16 | 2 |
| 84H05 | PF-LCre | 7 | 1 |
| 55G08 | Δ7 | 16 | |
| 47G08 | IS-P | 12 | |
| 49H05 | IMPL-F | 4 | |
| 75H04 | L-Ei | ~30 | |
| 32A11 | L-Em | ~24 | |
| 18A05 | GB-Eo | 2 | |
| 17H12 | AMPG-E | ~12 | |
| 12C11 | EFBG | 9 | |
| 72H06 | SMPL-L | ~12 | |
| SS02615 | SMPL-L2 | ~12 | |
| 26B07 | WL-L | 2 | |
| 31A11 | L-Cre | 2 | |
| SS00153 | S-P | 2 | |
| 76E11 | GL-N1 | 2 | |
| SS04448 | GL-N1 | 2 | |
| SS04420 | CreL-N2 | 2 | |
| 12G04 | L-N3 | 2 | |

DOI: https://doi.org/10.7554/eLife.37017.006

*Peña et al., 2014*; *Kahsai et al., 2012*; *Hanesch et al., 1989*). Note that the ring neurons presented here are non-canonical: they innervate the LAL or the Gall, but not the bulb. Second, a class of LAL-NO interneurons (the GL-N1 neuron) provides another source of inhibitory input into the EB columnar system by targeting the P-EN neurons (*Figure 5Aii* and *Figure 3Aii*). This connection is particularly interesting in the light of the finding that the P-EN neurons drive the rotation of the bump of activity in the heading representation system (*Green et al., 2017*; *Turner-Evans et al., 2017*). Since the left/right noduli segregation corresponds to individual cells coding turns in opposite direction (*Turner-Evans et al., 2017*), the GL-N1 neurons are likely involved in strengthening or refining those turn related signals. Moreover, it is likely that other types of LAL-NO interneurons innervating other noduli compartments target P-FN neurons, but these pairs have not been tested extensively yet.

We also identified two excitatory inputs to the central complex. First, a Gall to EB neuron, whose innervation pattern in the EB is reminiscent of the columnar neurons, excites the E-PG neurons (the same class that carries the heading representation and is inhibited by ring neurons). Second, several columnar neurons share excitatory inputs from the IS-P neuron in the PB (PB.b-IB.s-SPS.s, *Figure 5Aiii*). It is important to note that although we tested very few candidates in the FB, it is highly likely that this region receives many inputs from outside the central complex.

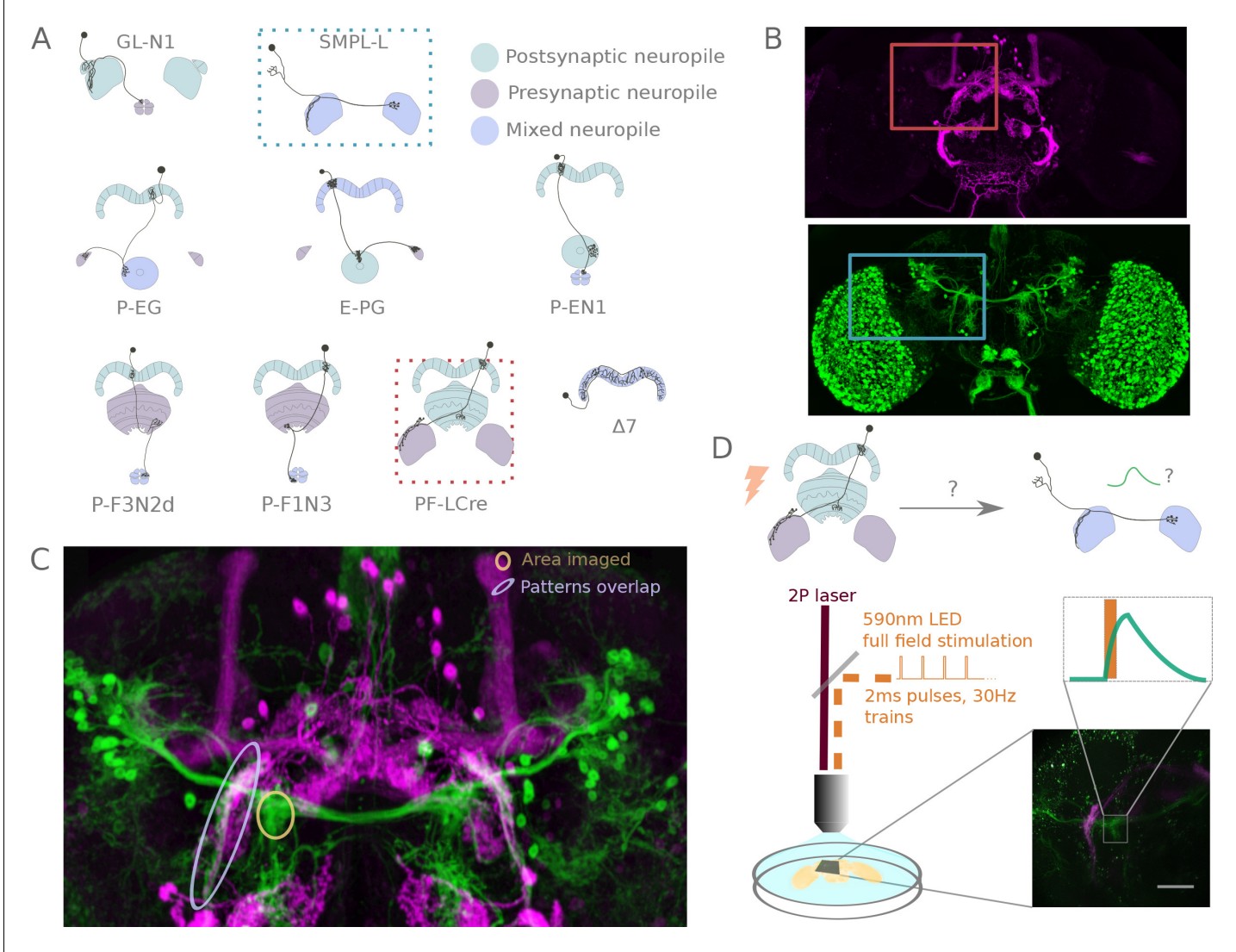

**Figure 2.** A functional connectivity screen. (**A**) Schematics of a subset of the neurons considered for the screen. The example neurons shown in B, C and D are indicated by blue and orange dotted-line boxes. (**B**) For each potential neuronal pair, driver lines with clean expression in the central complex were selected — the boxes delineate the approximate position of the neurons of interest in the brain. (**C**) To determine if a given pair is a promising candidate, we examined the degree of overlap between putative pre- and post-synaptic regions in the expression patterns in anatomy images registered to a common brain template. If the candidate pre- and post-synaptic regions overlapped (as indicated by the blue ellipse), we expressed CsChrimson in the presynaptic candidate and GCaMP6m in the postsynaptic candidate, and then imaged the ex-vivo brain in a two-photon laser scanning microscope while optogenetically stimulating the candidate presynaptic population (**D**). We selected the region imaged based on proximity to the overlapping processes, but ensured that it contained only GCaMP expressing arbors (yellow ellipse in C, and box in D).
DOI: https://doi.org/10.7554/eLife.37017.007

The neurons listed here do not necessarily provide feed-forward input from outside the central complex. For example, the gall ring neuron (GB-Eo), which is an inhibitory input to the E-PGs, likely participates in a feedback loop, as it receives excitatory input from P-EG neurons (*Figure 5Di*). It is possible that this kind of loop between the columnar system and input neurons from accessory structures is repeated at other places in the network. Another example would be the IMPL-F neuron, the FB-LAL neuron (top right corner of *Figure 4*) that receives input from the PF-LCre columnar neuron in the LAL. Its target in the central complex has not been identified so far, but since it is located in the FB, it likely involves the FB columnar system.

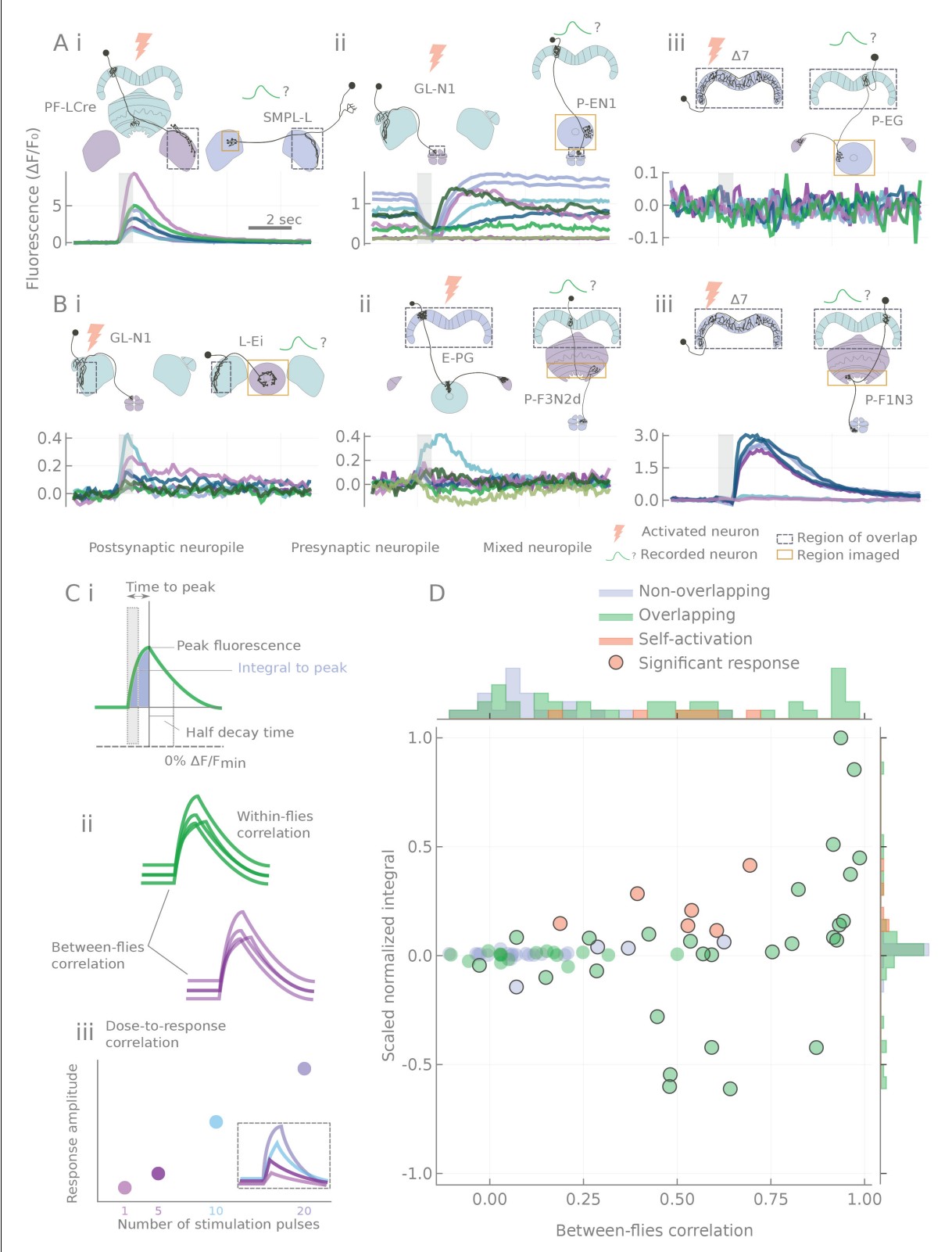

**Figure 3.** Characterization of calcium transients observed in response to o stimulation. (**A and B**) Summary of different response types. Stimulation is indicated by the gray bar and consists of 20 light pulses (50 µW/mm², each of 2 ms duration) delivered at 30 Hz. (**A**) Example neuron pairs, easily interpretable responses. (**B**) Example neuron pairs, responses that are more difficult to interpret. In A and B, all responses, expressed as $\Delta F/F_0$, are baseline subtracted except for the inhibitory response in Aii. Scale bar 2 s. Grey dashed boxes show the region of overlap between the two patterns,
*Figure 3 continued on next page*

*Figure 3 continued*

and yellow boxes indicate the area that was generally imaged for the pairs shown. (C) Example of statistics computed on individual runs and cell pairs characterizing: (i) average response shape, (ii) reliability of the response, and (iii) response sensitivity. (D) Using the distribution of statistics from non-overlapping controls to assign classes to the responses: distributions of response amplitudes and reliability as measured by the scaled normalized integral (the median of the integral normalized to the baseline and scaled so that the dataset spans the [−1,1] range) and the between-flies correlation (see Materials and methods). Each point corresponds to a different cell pair (median statistics across flies). Control unconnected pairs are shown in blue, and self-activation (CsChrimson and GCaMP6m expressed in the same neuron type) in orange. Responses considered significantly different from the control sample (p<0.01, see Materials and methods) are circled.

DOI: https://doi.org/10.7554/eLife.37017.008

The following video and figure supplements are available for figure 3:

**Figure supplement 1.** Responses of non-overlapping pairs.

DOI: https://doi.org/10.7554/eLife.37017.009

**Figure supplement 2.** Distributions of the measured statistics for the different groups.

DOI: https://doi.org/10.7554/eLife.37017.010

**Figure supplement 3.** Effects of baseline fluorescence fluctuations on responses.

DOI: https://doi.org/10.7554/eLife.37017.011

**Figure 3—video 1.** PF-LCre to SMPL-L response, also shown in Ai.

DOI: https://doi.org/10.7554/eLife.37017.012

**Figure 3–video 2.** GL-N1 to P-EN1 response, also shown in Aii.

DOI: https://doi.org/10.7554/eLife.37017.013

**Figure 3–video 3.** E-PG to PF-LCre response, not shown in figure.

DOI: https://doi.org/10.7554/eLife.37017.014

## Outputs

In contrast with inputs, we found few potential channels leaving the central complex. The only output pathway identified in this dataset is presented in *Figure 5C*, and connects the PF-LCre columnar neuron to a LAL interneuron through a strong, mecamylamine-sensitive (hence cholinergic, see Materials and methods), excitatory connection (see *Figure 3Ai* and *Figure 4—figure supplement 2* for the pharmacology). This information is likely further processed in the LAL, as we found indications of inhibition upon PF-LCre stimulation in another LAL interneuron (WL-L). Even if we could not trace the circuit responsible for this inhibition, it likely involves an intermediate interneuron in the LAL. Once again, as this dataset does not include every single cell type of the central complex, some outputs might easily have been missed. FB tangential neurons (*Hanesch et al., 1989*), for example, may also contribute output pathways.

## Connectivity in the protocerebral bridge

A functional connectome is, by construction, sparser than can be predicted by light-level anatomy. Our study shows this most clearly in one neuropil, the PB (*Figure 5B*). E-PG neurons are the only columnar type that are presynaptic in the PB, but their activation did not trigger a significant response in any of the five other columnar neurons we tested. This came as a surprise because we assumed that the E-PG population would connect to the rest of the EB and FB columnar systems. To verify that this lack of observed connectivity was not due to the recruitment of global inhibitory circuits, we also ran these experiments in the presence of picrotoxin, and did not observe any difference in responses (see *Figure 4—figure supplement 3*). By contrast, the Δ7 interneurons are strongly activated by E-PG neurons, and their activation leads to significant responses in several columnar neuron types (E-PG, P-EN1, P-EN2, P-F1N3 and P-F3N2v). The Δ7 neurons, therefore, appear to constitute an important bottleneck in the system (*Figure 5B*), and may serve as the only strong link between columnar neurons in the PB. The response profiles following Δ7 activation are also unusually complex (see *Figure 5—figure supplement 1*): P-ENs display mild activation, E-PG and P-F3N2v inhibition, and P-F1N3 strong rebound excitation (see *Figure 3Biii*).

## Connectivity in the EB columnar system, the ring attractor network

*Figure 5Dii* shows the subpart of the network that has been proposed to sustain the ring attractor representation of heading (*Green et al., 2017*; *Turner-Evans et al., 2017*). One hypothesized feature of such a circuit is a large degree of recurrence between the different EB columnar types. In

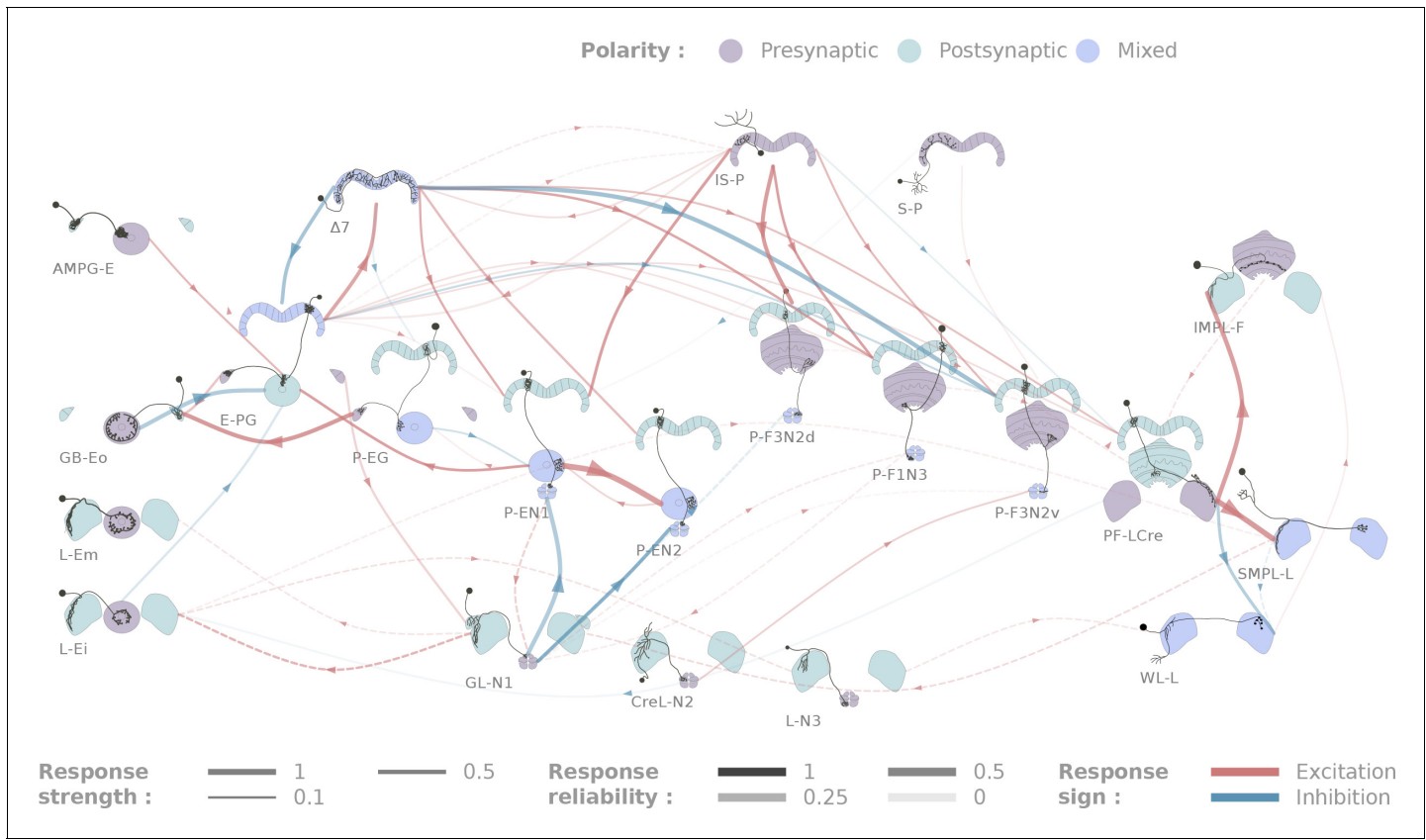

**Figure 4.** Diagrammatic representation of central complex connectivity. Solid lines indicate anatomically overlapping cell pairs, whereas dotted lines correspond to the non-overlapping controls. The thickness of the lines maps to the functional connection strength. The reliability of the responses as measured by the between-flies correlations is mapped to the transparency of the connectors. Connection strengths are quantified in terms of the Mahalanobis distance to the null sample and the sign of the response integral, which is normalized to the maximum response. Three tested cell types (SMP-L2, L-Cre and EFBG) are not shown here for clarity, as tests with a few different partners did not produce significant results.

DOI: https://doi.org/10.7554/eLife.37017.015

The following figure supplements are available for figure 4:

**Figure supplement 1.** Matrix representation of the same data.
DOI: https://doi.org/10.7554/eLife.37017.016
**Figure supplement 2.** Results of mecamylamine applications.
DOI: https://doi.org/10.7554/eLife.37017.017
**Figure supplement 3.** Results of picrotoxin applications.
DOI: https://doi.org/10.7554/eLife.37017.018
**Figure supplement 4.** Dose response curves.
DOI: https://doi.org/10.7554/eLife.37017.019
**Figure supplement 5.** Responses as a function of genotype used.
DOI: https://doi.org/10.7554/eLife.37017.020

particular, P-EN to E-PG reciprocal connections are important for models of the rotation of the bump. While we found strong support for the P-EN1 to E-PG connection, the E-PG to P-EN1 connection that we reported functionally under a stronger stimulation protocol (*Turner-Evans et al., 2017*) may be mediated through the Δ7 interneurons. A few other connections were found in the EB (for example, P-EN1 to P-EN2), but it is important to stress that not all combinations could be tested due to limitations in the genetic reagents available. For example, the role of the P-EG neurons in this circuit, remains unclear. A key additional type that our results suggest may contribute in important ways to the persistence of activity in this circuit is the AMPG-E neuron, a columnar Gall-EB neuron not innervating the PB, which appears to provide localized excitatory feedback to the E-PG neurons.

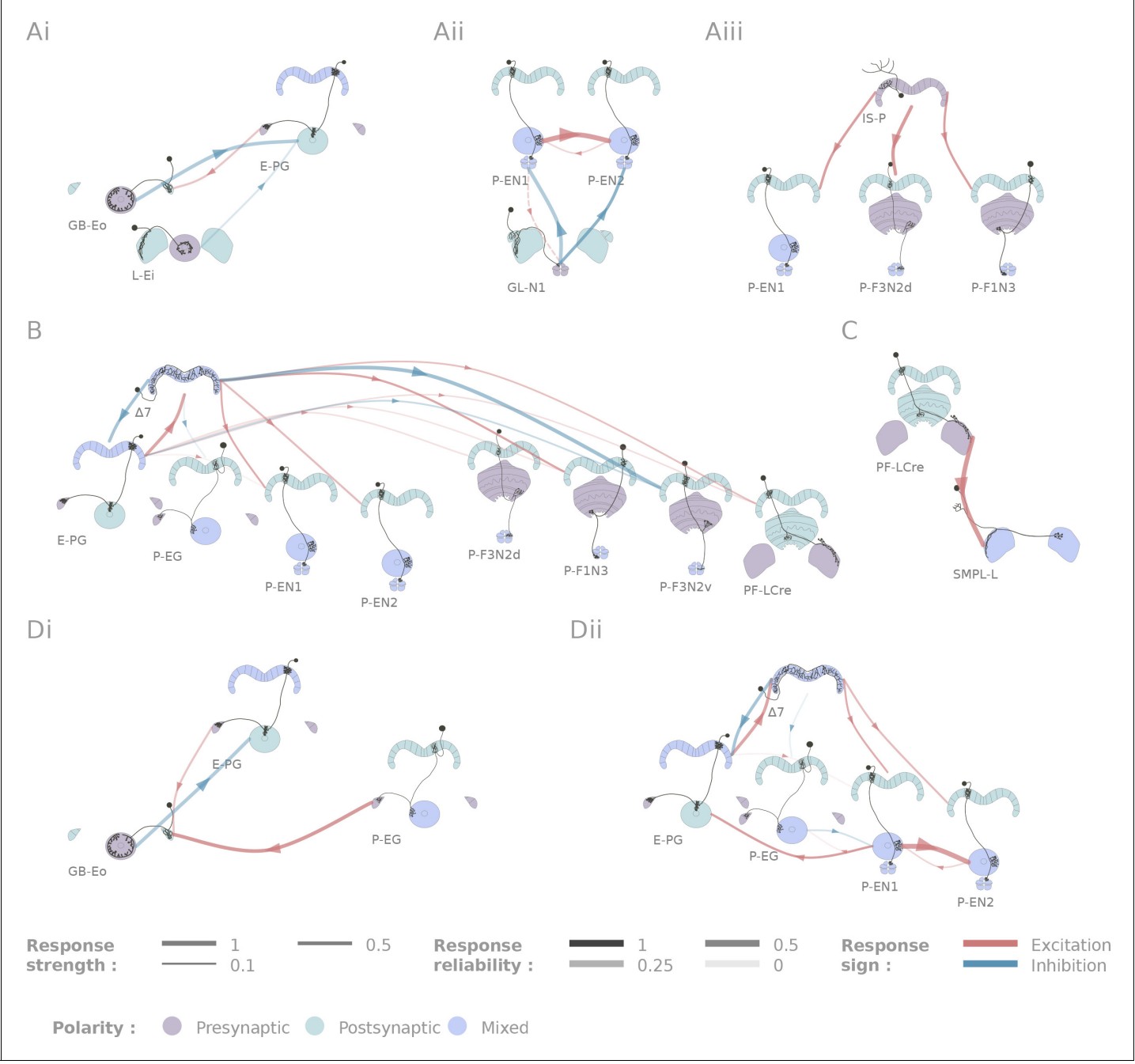

**Figure 5.** Selected connectivity motifs within the central complex. (**A**) Input channels. (**i**) Ring neurons provide inhibitory input to the E-PGs in the EB. (**ii**) GL-N1 inhibits P-EN neurons. (**iii**) Distributed excitatory input from IS-P neurons in the PB. (**B**) Δ7 is probably the bottleneck in PB motifs, as it is the only strong post-synaptic target of E-PG neurons and relays information to other columnar neurons. (**C**) The only output pair found so far connects the PF-LCre neuron to a LAL interneuron. (**D**) Recurrence in the central complex. (**i**) at the input stage, and (**ii**) within the EB columnar system.

DOI: https://doi.org/10.7554/eLife.37017.021

The following figure supplement is available for figure 5:

**Figure supplement 1.** Various response types following Δ7 stimulation.

DOI: https://doi.org/10.7554/eLife.37017.022

## Discussion

The dataset presented in this study constitutes a resource for the growing community of researchers interested in the central complex. While similar coarse functional connectivity techniques have been used to map short pathways in previous studies, this is, to our knowledge, the first extensive dataset of its kind. We hope that it will become an evolving source of information, which we expect to be most useful when combined with other complementary data sources, such as EM-level anatomical connectivity and high-resolution gene expression profiles. Such combined data would constitute a solid base to build constrained network models of the central complex, and to generate detailed hypotheses of its function. As with any large dataset, we see this effort mainly as a starting point for more detailed research.

### Limitations of the method

The connectivity technique we applied has several limitations that are important to keep in mind. First, connections detected using CsChrimson and GCaMP cannot be guaranteed to be direct and monosynaptic. However, the large set of controls with cell-type pairs whose processes do not overlap provides a statistical framework to interpret the results — not surprisingly, uncertainty is highest for weak connections. We believe that the metric we used to assess connectivity – distance to a control set rather than just response strength – makes the resulting network more interpretable. Additionally, by releasing the entire dataset rather than just the derived network, we hope to provide interested central complex researchers the opportunity to explore the data and to potentially reinterpret it in the light of other findings.

A more fundamental issue concerns the sensitivity of our protocol, which is limited by the stimulation protocol and the sensitivity of GCaMP6m. Specifically, an absence of a post-synaptic response cannot be interpreted as an absence of a connection. The fact that some inhibitory responses are visible, and that strong responses saturate with the range of stimulations used (see *Figure 4—figure supplement 4*) is reassuring. However, it is likely that EM reconstructions of central complex circuits will reveal that some weak synaptic connections have been missed by our technique. Their functional importance will need to be investigated using more sensitive methods, for example, intracellular electrophysiology. For example, we may be underestimating the level of connectivity in the PB: our finding of sparseness in the structure should thus be interpreted as sparseness at the resolution of our technique, because this network may be dominated by weak connections below our detection threshold.

Further, we relied on full-field stimulation of populations of specific neuronal types, which comes with its own drawbacks: this approach provides no access to connectivity between neurons of the same class, and does not account for potential non-physiological network effects. One such effect would be the recruitment of global inhibitory networks that could mask an otherwise excitatory connection. However, whenever we suspected this could be a possibility, we controlled for it by blocking inhibition with picrotoxin, and never saw evidence of a significant effect (*Figure 4—figure supplement 3*). Even though picrotoxin was effective in blocking the inhibitory responses we observed when activating ring neurons or Δ7 neurons (*Figure 4—figure supplement 3*), we cannot exclude the possibility that picrotoxin-insensitive inhibition might be present in the network. Possible candidates mediating such effect would be GABA-B (*Olsen and Wilson, 2008*), metabotropic glutamatergic or peptidergic transmission (*Kahsai and Winther, 2011*). For example, the PB shows both GABA-B and metabotropic glutamatergic receptor immunoreactivity (*Kahsai et al., 2012*), which could make the observed connectivity seem sparser than it actually is. Interestingly, whereas in other insect species the PB displays peptidergic immunoreactivity (in particular Allatostatin-A, see [*Vitzthum et al., 1996*]), this seems not to be the case in *Drosophila* (with the exception of SIFamide, see [*Kahsai et al., 2012*]).

The fact that we stimulate entire presynaptic populations also means that the strength of connections we report is influenced both by neuron-to-neuron transmission strength and the degree of convergence in the network. Caution should therefore be exercised when comparing connections between columnar neuron population and other types (e.g. from PF-L and SMPL-L or P-EG to G-Eo), which are potentially highly convergent, to connections between columnar neuron types, or from accessory structure neuron types to columnar populations (like GL-N1 to P-EN1), which are both mediated by a single or small handful of presynaptic neurons for any given post-synaptic neuron.

Furthermore, even though several of the neurons described have complex morphologies that are suggestive of local processing within specific neuropiles, we never found neuropile-specific responses: when a neuron responded, the response seemed to invade the entire neuron. This may be the result of our broad and artificial stimulation protocol.

Given that our protocol is limited to the activation of one cell type, we might also have missed connections gated by other inputs. For example, we failed to find any PB input for the PF-LCre neurons – the sole output neurons we identified in this study. It is possible that this neuron requires convergent inputs from the PB and FB to be activated, as was suggested in (*Stone et al., 2017*).

Finally, all our experiments were performed in ex vivo brain preparations. Given the variety of neuromodulators that operate in the central complex (*Kahsai and Winther, 2011*), it is likely that functional connectivity within this region is modulated by brain state (*Homberg, 1994*; *Seelig and Jayaraman, 2013*; *Weir et al., 2014*; *Weir and Dickinson, 2015*). Consistent with this possibility, we saw that the fluorescence baseline tended to fluctuate spontaneously during the course of our experiments in most types recorded (as shown in *Figure 3—figure supplement 3A*). Two neuron types had relatively small baseline fluctuations that could have make it difficult to detect inhibition when those were used as post-synaptic targets: P-EG and PF-LCre. Intriguingly, although increases in baseline activity allowed us to detect inhibitory responses, we noticed that excitatory responses also occasionally depended on this baseline fluctuation (*Figure 3—figure supplement 3C*). It is conceivable that such baseline fluctuations reflect a kind of artificial brain state upon which the response amplitudes depend.

## Potential neurotransmitters

From our pharmacology experiments, we propose that columnar neurons and IS-P neurons are likely cholinergic (see *Figure 4—figure supplement 2*), whereas the LAL and Gall ring neurons, as well as the Δ7 neurons are either glutamatergic or GABAergic (see *Figure 4—figure supplement 3*). A large fraction of 'canonical' BU-ring neurons have been shown previously to be GABAergic (*Zhang et al., 2013*), which makes this likely for the LAL and Gall-ring neurons described here. We argue below that the response profiles of Δ7 neurons suggests that they are glutamatergic, in accordance with (*Daniels et al., 2008*).

## Central complex motifs

The connectivity matrix we obtained is sparser than that predicted by light level anatomy. Our results suggest that the Δ7 interneurons are a bottleneck for information processing in the PB. This is all the more interesting given the range of responses evoked by Δ7 stimulation (*Figure 5—figure supplement 1*). Properties of the synapses that Δ7 neurons make with their post-synaptic partners may play a primary role in the way that a heading signal is generated and maintained in the EB columnar system, and also in how it may be transferred to the FB columnar system. This has also been suggested in other insect species (*Stone et al., 2017*) for the homologous TB1 neuron, which was a key component of the proposed compass circuit model in that study. Every Δ7 neuron innervates all columns of the PB, and has presynaptic-looking processes in two columns. The fact that a neuron with such extensive arbors participates in a circuit where representations are spatially restricted (heading-related activity is limited to a few neighboring columns at any given time) suggests that understanding local processing at the single neuron level might be critical to a complete understanding of how the circuit as a whole operates. This may also be the case for some of the ring neurons that provide input to the ellipsoid body.

The observation that Δ7 neuron stimulation can excite or inhibit its post-synaptic partners can have several explanations. Either the population of Δ7 is not homogeneous, and contain several functionally distinct types responsible for the different types, or the responses reflect a variety of receptors on the post-synaptic side. This latter hypothesis would be compatible with the previously discussed hypothesis that Δ7 are glutamatergic, as the diversity of ionotropic and metabotropic glutamate receptors would allow such response diversity.

The fact that several sources of input are inhibitory raises the question of how activity is maintained in the region. Candidate mechanisms are the uncovered excitatory inputs into the PB and EB, recurrent connections in the EB and intrinsic properties of neurons (*Egorov et al., 2002*; *Yoshida and Hasselmo, 2009*; *Russell and Hartline, 1982*) — some cell types, for example, showed

robust post-stimulation rebounds (see *Figure 3Bii*). It is also possible that our selection of cell types and our methods missed some sources of excitation.

The range of inputs revealed here opens many avenues for investigation. Whereas some ring neuron subtypes have received considerable attention (*Sun et al., 2017*; *Shiozaki and Kazama, 2017*; *Seelig and Jayaraman, 2013*), most PB inputs and LAL-noduli interneurons have not yet been characterized. A recent study in the sweat bee (*Stone et al., 2017*), for example, reported that one of the LAL-noduli interneurons — a likely input to the FB system — carries forward and backward translational optic flow signals. This is all the more interesting given that we show that one of the LAL-NO interneuron types (GL-N1) provides input to the P-EN neurons, known to encode rotational signals (*Turner-Evans et al., 2017*; *Green et al., 2017*).

The specific functions subserved by the network motifs that we have uncovered may only become clear with functional studies in behaving animals. A key puzzle set up by our findings is the small number of output channels of the central complex. Our results are consistent with the LAL being the primary output structure for the central complex (*Chiang et al., 2011*; *Hanesch et al., 1989*), although the structure also acts as an input region (via ring neurons and potentially via IMPF-L neurons). While it is possible that our selection of Gal4 lines was unintentionally biased against output neurons, or that our technique otherwise missed a number of output pathways, the picture of the central complex that emerges is of a densely recurrent sensorimotor hub with relatively low dimensional output (much as proposed by some models for example [*Stone et al., 2017*; *Fiore et al., 2015*; *Strauss and Berg, 2010*]). The implications of this bottleneck for motor control remains a challenge for future studies to resolve.

## Materials and methods

### Fly stocks and crosses

Drivers were chosen based on relatively sparse expression within the central complex. For any given pair of neurons, the overlap between pre- and post-synaptic looking regions was assessed based on publicly available expression patterns ([*Tirian and Dickson, 2017*; *Jenett et al., 2012*], see *Figure 1—figure supplement 1*) digitally aligned on a common reference brain (as described in [*Aso et al., 2014*]). For every LexA driver used, we prepared two stocks containing GCaMP6m (*Chen et al., 2013*) and CsChrimson (*Klapoetke et al., 2014*) under LexAop (resp. UAS) or UAS (resp. LexAop) control: *XXX-LexA;13XLexAop2-IVS-p10-GCaMP6m 50.629 in VK00005, 20xUAS-CsChrimson-mCherry-trafficked in us(How)attP1* and *XXX-LexA;20xUAS-IVS-GCaMP6m 15.629 in attP2, 13XLexAop2-CsChrimson-tdTomato in VK00005*. Those stocks were then crossed to a Gal4 driver or a split-Gal4 (*Luan et al., 2006*) driver for the experiment. For split-Gal4s, the two split halves were inserted in attP40 and attP2 respectively. To avoid transection between the split and the LexA driver (*Mellert and Truman, 2012*), we inserted the LexA drivers in alternative sites, either su(Hw)attP5 (*Pfeiffer et al., 2010*) or VK22 (*Venken et al., 2006*), and used the splits exclusively in combination with those lines after checking their expression patterns. The list of drivers used and the corresponding cell types are given in *Table 1*. Throughout this paper, we follow the naming convention set out in *Wolff et al. (2015)* for full names, and the scheme described in *Kakaria and de Bivort (2017)* and used in *Green et al., (2017)* and *Turner-Evans et al., (2017)* for abbreviations. For each cell type, we labeled every region innervated as presynaptic or postsynaptic (or both): this was done at the resolution of the glomerulus for the PB, the layer for the FB and the individual nodulus. We divided the LAL into three zones based on the overlap between the lines used. Existing subdivisions for the EB and Gall were preserved. This labeling was used to evaluate whether the arbors of a given cell-type-pair overlapped.

### Dissections

The brains of 5 to 9 days old female flies were extracted and laid on a poly-D-lysine coated coverslip (Corning, Corning, NY). In most experiments, both the brain and the ventral nerve chord (VNC) were dissected out, as we found that having the VNC attached to the brain increased the mechanical stability of the preparation. Dissection was performed using the minimum level of illumination possible to avoid spurious activation of CsChrimson. The preparation was bathed throughout in saline containing (in mM): 103 NaCl, 3 KCl, 5 TES, 8 trehalose dihydrate, 10 glucose, 26 NaHCO3, 1 NaH$_2$PO$_4$,

2 CaCl$_2$, 4 MgCl$_2$, bubbled with carbogen (95% O$_2$, 5% CO$_2$). Brains were positioned anterior-side-up, except when the connection tested was thought to be in the PB, in which case they were positioned posterior-side-up to maximize light access close to the assumed synaptic site. Trachea were removed. Only for experiments involving pharmacology, the glial sheath was gently torn with tweezers to enhance drug access to the neuropiles.

## Imaging conditions and trial structure

Imaging was performed on an Ultima II two-photon scanning microscope (Bruker, Billerica, MA) with a Vision II laser (Coherent, Santa Clara, CA). Brains were continuously perfused in the saline used for dissection at 60 mL/hr. Once the sample was placed and centered under the objective, we waited 5 min before starting the experiment to avoid any lingering network activation from the dissection or transmission lights. Two-photon excitation wavelength was 920 nm, and power at the sample varied between 3 and 10 mW. CsChrimson was excited with trains of 2 ms long 590 nm light pulses via an LED (M590L3-C1, Thorlabs, Newton, NJ) shone through the objective. The excitation light path contained a 605/55 nm bandpass filter and was delivered to the objective with a custom dichroic (zt488-568tpc, reflecting between 568 nm and 700 nm). A 575 nm dichroic beam splitter and bandpass filters (525/70 nm and 607/45 nm for the green and red respectively) were placed in the detection arm before photons reached the PMTs (Hamamatsu multi-alkali). Instantaneous power measured out of the objective was roughly 50 µW/mm$^2$. Stimulus pulse trains were delivered at 30 Hz and the number of pulses varied between 1, 5, 10, 20 and 30 — corresponding to total stimulation durations ranging from 2 ms to 1 s. Imaging fields of view were chosen as to avoid scanning regions containing CsChrimson-expressing neuropil while being as close as possible to the putative connection site, as we observed occasional two-photon-evoked slow activation of CsChrimson-expressing cells (high-intensity two-photon stimulation of CsChrimson was used for spatially precise neuronal activation in [*Kim et al., 2017*]). When this was impossible — for example, in self-activation controls or for completely overlapping cell types — we chose a large ROI of which the CsChrimson/GCaMP6m-expressing neuropile represented a small fraction, so as to minimize duty cycle. ROIs were kept constant throughout the experiment. Each experimental run consisted of four repeats, each approximately 16s long. Runs were themselves repeated every 2 min. All experiments started with five runs corresponding to the five stimulation strengths, in a random order. This was sometimes followed by pharmacological testing. At the end of the experiment, a high intensity 3D stack was acquired to check that the expression patterns were as expected, and that the neutrophil imaged was the targeted one in cases where fluorescence levels during the experiments were very low. At least six flies were tested for every pair considered.

## Pharmacology

For blocking nicotinergic or inhibitory (GABAergic or glutamatergic) transmission, mecamylamine (50 µM) or picrotoxin (10 µM) (Sigma-Aldrich, St Louis, MO) were perfused by switching to a different line for 3 min, followed by a wash period during which the perfusion was drug-free again. Thirty pulses stimulation runs were repeated every 2 min, starting 4 min before the drug application and throughout the wash. Prior to use, solutions were kept frozen in 25 mM and 0.3 M aliquots, respectively.

## Analysis

All analyses were performed in http://julialang.org/Julia, using custom-written routines. All data and code are available as an OpenScienceFramework project at https://osf.io/vsa3z/ (*Franconville, 2018a*). Code is also centralized in a Github repository ([*Franconville, 2018b*], https://github.com/romainFr/CX-Functional-Analysis; copy archived at https://github.com/elifesciences-publications/CX-Functional-Analysis) and notebooks recapitulating the analysis can be run directly from the browser at https://mybinder.org/v2/gh/romainFr/CX-Functional-Analysis/master (using https://mybinder.org/Binder).

### Data processing

For a given experiment, all movies were aligned to each other to compensate for slow drifts of the sample: for each run, the average image was calculated, and translation drifts between average

images were calculated using correlation-based sub-pixel registration ([*Guizar-Sicairos et al., 2008*], and https://github.com/romainFr/SubpixelRegistration.jl for the Julia implementation used here; copy archived at https://github.com/elifesciences-publications/SubpixelRegistration.jl). A region of interest (ROI) was defined for the full experiment: the average image (of all the runs) between foreground and background was distinguished using k-means clustering. Note that the selection method relies only on average intensity and not activity —a method we chose so as to maintain the same detection method for responsive and non-responsive runs. This also relies on selecting fields of view as unambiguously containing the neuron of interest — and only the neuron of interest — during the experiment.

$\Delta F/F_0 = \frac{(F-F_0)}{(F_0-B)}$, where F is the raw fluorescence and B the background signal (calculated as the intensity of the dimmest 10% pixels of the average image) were then computed for each movie in the ROI. Given that baseline fluorescence could vary widely over the course of an experiment (see Discussion), we defined $F_0$ as the median fluorescence in the ROI in the dimmest 3% of frames of the entire experiment.

## Statistics

For every experimental repeat, we computed the following statistics:

- $F_{peak}$, the peak fluorescence value, and $T_{peak}$, the time after stimulation at which the peak value is reached
- $I_{toPeak}$, the integral of the signal from the onset of stimulation to the peak time
- $\tau_{1/2}$, the half-decay time of the fluorescence signal after the peak is reached
- $F_{base}$, the fluorescence baseline before stimulation expressed in $\Delta F/F_0$
- $F_{peak\_norm}$ and $I_{toPeak\_norm}$, the fluorescence peak and integral normalized by the baseline: $F_{peak\_norm} = F_{peak}/F_{base}$ and $I_{peak\_norm} = I_{peak}/F_{base}$(see Discussion)

Then, for every run, which consists of 4 repeats done on the same fly, we computed:

- $R_{within\_flies}$, the average correlation between the four repeats of the fluorescent transient
- $<F_{peak}>$, $<T_{peak}>$, $<I_{toPeak}>$, $<F_{base}>$, $<F_{peak\_norm}>$, $<I_{to\,peak\_norm}>$, and $<\tau_{1/2}>$, the medians across the 4 runs of $F_{peak}$, $T_{peak}$, $I_{toPeak}$, $F_{base}$, $F_{peak\_norm}$, $I_{toPeak\_norm}$, and $\tau_{1/2}$, respectively.

Subsequently, for every set of runs done on the same cell pair and the same stimulation protocol, we computed:

- $<<F_{peak}>>$, $<<T_{peak}>>$, $<<I_{toPeak}>>$, $<<F_{base}>>$, $<<F_{peak\_norm}>>$, $<<I_{toPeak\_norm}>>$, $<R_{within-flies}>$ and $<<\tau_{1/2}>>$, the medians across flies of $<F_{peak}>$, $<T_{peak}>$, $<I_{toPeak}>$, $<F_{base}>$,$<F_{peak\_norm}>$, $<I_{toPeak\_norm}>$, $R_{within-flies}$ and $<\tau_{1/2}>$, respectively
- $R_{between-flies}$, the average correlation between the average fluorescent transients of individual runs
- $R_{state}$, the correlation between $I_{toPeak}$ and $F_{base}$ across flies

Moreover, we created $<I_{toPeak}>_{scaled}$, $<<I_{toPeak}>>_{scaled}$, $<I_{toPeak\_norm}>_{scaled}$ and $<<I_{toPeak\_norm}>>_{scaled}$, scaled versions of $<I_{toPeak}>$, $<<I_{toPeak}>>$, $<I_{toPeak\_norm}>$ and $<<I_{toPeak\_norm}>>$ so that the values cover the range $[-1,1]$ by scaling positive (negative) values by the maximum (minimum) response in the dataset.

## Distance from control and significance

Based on light level anatomy, we labeled each tested pair as overlapping or non-overlapping. We used the set of non-overlapping pairs as a control (the null sample). Considering only two parameters, the scaled normalized integral $<<I_{toPeak\_norm}>>_{scaled}$ and the correlation across flies $R_{between-flies}$ (see *Figure 3*), we calculated the Mahalanobis distance between the null sample and each data point, using a robust estimate of the covariance matrix (following [*Rousseeuw and Driessen, 1999*]) of the null sample. For a point $\vec{x} = (<<I_{to\,peak\_norm}>>_{scaled}, R_{between-flies})^T$ this will be $\sqrt{(\vec{x}-\vec{\mu})^T S^{-1} (\vec{x}-\vec{\mu})}$ where $S$ is the covariance matrix and $\vec{\mu}$ is the vector the average parameters for the null sample. While single statistics never were sufficient to capture all relevant aspects of the response, we found that these two measurements recapitulated well distance measurements

obtained by combining all the statistics. We then computed 99% confidence intervals on the distribution of distances by bootstrapping to determine significance.

## Preprint

An earlier version of this manuscript is available as a preprint at https://www.authorea.com/155729/_TsHpd9reMuWijjossgt6Q (DOI: 10.22541/au.151537454.41878908).

# Additional information

## Funding

| Funder | Author |
| --- | --- |
| Howard Hughes Medical Institute | Romain Franconville<br>Celia Beron<br>Vivek Jayaraman |

The funders had no role in study design, data collection and interpretation, or the decision to submit the work for publication.

## Author contributions

Romain Franconville, Conceptualization, Data curation, Software, Formal analysis, Validation, Investigation, Visualization, Methodology, Writing—original draft, Writing—review and editing; Celia Beron, Validation, Investigation; Vivek Jayaraman, Conceptualization, Supervision, Funding acquisition, Project administration, Writing—review and editing

## Author ORCIDs

Romain Franconville (iD) http://orcid.org/0000-0002-4440-7297
Vivek Jayaraman (iD) http://orcid.org/0000-0003-3680-7378

## Decision letter and Author response

Decision letter https://doi.org/10.7554/eLife.37017.029
Author response https://doi.org/10.7554/eLife.37017.030

# Additional files

## Supplementary files

• Supplementary file 1. Summary table of experiments. The table contains condensed information about the individual experiments: Cell Pair: the cell pair tested, genotype: the LexA, Gal4 and expression system driving CsChrimson (enough to reconstruct the full genotype, see Materials and methods, region imaged: the neuropile imaged for a given experiment. in or out suffixes denote that the processes entering (exiting) the neuropile were imaged, data folder: the raw data folder corresponding to that experiment in the OpenScienceFramework project (https://osf.io/vsa3z/), Stack: the number corresponding to the anatomical stack for that experiment in the previously identified data folder.
DOI: https://doi.org/10.7554/eLife.37017.024
• Transparent reporting form
DOI: https://doi.org/10.7554/eLife.37017.025

## Data availability

All the data generated or analyzed during this study are freely available for exploration at: https://romainfr.github.io/CX-Functional-Website/ All code and data are available at: https://osf.io/vsa3z/

The following dataset was generated:

| Author(s) | Year | Dataset title | Dataset URL | Database, license, and accessibility information |
|---|---|---|---|---|
| Franconville R | 2018 | Central complex functional connectivity | https://osf.io/vsa3z/ | Publicly available at the Open Science Framework (DOI 10.17605/OSF.IO/VSA3Z \| ARK c7605/osf.io/vsa3z) |

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
