## [Decision Letter]

Thank you for submitting your article "Building a functional connectome of the *Drosophila* central complex" for consideration by *eLife*. Your article has been reviewed by K VijayRaghavan as the Senior Editor, a Reviewing Editor, and three reviewers, Kristin Scott as the Reviewing Editor, and three reviewers. The following individual involved in review of your submission has agreed to reveal his identity: Stanley Heinze (Reviewer #1).

The reviewers have discussed the reviews with one another and the Reviewing Editor has drafted this decision to help you prepare a revised submission.

Summary:

The paper 'Building a functional connectome of the *Drosophila* central complex' by Franconville et al. is an impressive and important piece of work. It shows, for the first time, functional connections within the insect central complex (CX), a region that has gained substantial attention over the last decade and is of fundamental importance to understanding action selection in insects. The authors present a large dataset of functional connectivity in the central complex and accessory structures, collected from testing pairs of cell types expressing optogenetic activator and genetically encoded calcium indicator. The paper not only reveals connections that previously had to be indirectly inferred, it also provides a road map of how to expand this work into a comprehensive map of connectivity of this region, a method that will likely also be applicable for other brain areas. The insect central complex is an interesting and appropriate target for such an impressive approach and the study confirms previous hypotheses about information flow.

Essential revisions:

1) The title overstates what is shown in the paper. The claim of a functional connectome raises the expectation that, if not all neurons, then at least all major neurons should be included in the analysis. Neither is the case. First, the main and best described input cells to the CX, the ring neurons connecting the bulbs to the ellipsoid body, are not included. Second, the pontine cells, numerically one of the most common neurons in the locust CX, are entirely missing from the study. Third, the input cells to the fan shaped body were largely excluded from the work as well. An introduction that better describes the anatomy of the CX and the cells examined in this study and a title that does not claim to build a functional connectome would improve this work.

2) Many columnar neurons of the CX exist in more than one individual cell per column. Most extremely this was recently shown in bees, where 18 CPU4 cells (= P-FN cells) were shown to coexist in each column (Stone et al., 2017). This gives rise to the possibility that these cells could form circuits with each other rather than only with other types of neurons. As this appears to be a theme repeated for many cell types of the CX, this adds an entire layer of complexity to the CX-connectome that is left unexplored in the current paper. As any given driver line might only label a subset of anatomically identical cells (as shown for the P-EN cells), it would be key for interpreting the data, if the authors could provide information as to how many neurons per column are included in their drivers (to get an idea of how many cells were missed, once the total numbers for each cell type become available via other means).

3) The stimulation protocol only allows activation of all cells in a given population (rather than individual cells). As one hallmark of CX neurons is their spatially distinct activity across each population, this is a highly unnatural activity pattern, that can be expected to mask patterns of postsynaptic recruitment. The authors do mention this in the discussion and address the issue of recruiting global inhibitory circuits. The control for this with picrotoxin is an essential step in the right direction, but the possibility of non-GABAergic inhibition is looming in the background. This is especially important as the emphasized bottleneck of the CX circuit (the Delta7 cells) have been described as peptidergic (Allatostatin expressing) in other insects (TB1 cells are the homologous cells in all other insects). Inhibition mediated by this peptide has recently been shown to play an important role in mediating sleep in the CX of *Drosophila* (10.1016/j.neuron.2017.12.016). The conclusions drawn from the lack of direct columnar neuron connections within the PB are therefore to be taken with caution and needs more extensive discussion.

4) Even though the focus of the work lies on *Drosophila*, much of its importance results from the broad impact of the paper for other insects. The discussion should be modified to place the work in context with studies from other insects. There is a lot known about the neurons constituting the described input and output pathways of the CX in locusts, bees, beetles and butterflies. Also, the homologous neuron of Delta7 (TB1) has been known to play a key role in the compass circuit of locusts for a decade and has been described in much detail (e.g. 10.1002/cne.23736, 10.1126/science.1135531). In fact, the described connectivity (indirect link between E-PG and columnar output cells via Delta7) in the current paper was also a prediction of Stone et al., (2017) in bees. There, this neuron was instrumental in constructing a ring attractor model needed to generate a compass signal that was required as direction signal for a path integration circuit. While some differences exist between the detailed TB1/Delta7 morphologies, the overall pattern that defines the cell is highly conserved across all species. Similarly, the output cells shown as being the only CX-output in the current work, are equivalent to the CPU1 cells in other insects, which have been proposed to be the sole output of the compass circuit (or path integration circuit model) in those species as well.

5) The CX shows state dependent neural activity, is involved in sleep/attention regulation, and is predicted to drive actions in a context dependent way. Thus, output from this region will almost surely depend on the behavioral and motivational state of the animal, which is not maintained in the ex-vivo preparation used in this paper. Synaptic transmission efficiency can be expected to be differentially modulated in different behavioral contexts when several inputs converge onto the output cells (as suggested e.g. by Stone et al., 2017). With this in mind, the finding that there is no strong input from any cell onto the only output cell of the system needs to be discussed. Similarly, the baseline fluctuations that allowed the authors to identify inhibitory connections are likely a result of state-changes of the neural circuit. Yet it is unclear whether these occur in all neurons equally likely. The authors indeed hint that this might not be the case, so that it appears unclear as to how many inhibitory connections could have been overlooked. The ex-vivo nature of the experiments should be mentioned early in the paper and the potential implications of this rather unnatural state of the brain should be discussed more clearly.

6) Cell types and cell pairs tested: There is mention of 24 cell types, but it seems only 21 types were tested. It is also unclear which percentage of potential synaptic partners were tested- are there neurons which overlap in their arbors, but were not yet tested? In addition, is it possible to roughly estimate the percentage of central complex neurons covered by this screen? If yes, the authors should include this information in the paper.

7) A detailed description of each tested pair with the genotype is necessary and could be included as a table. Information about the exact Gal4-LexA pairs that are tested, the full genotypes of each tested pair, the pre and post synaptic cell names attached, the neuropils each cell occupies, and the observed connectivity would provide a great "cheat sheet". In addition, image stacks for the tested pairs should be made available online, if not already available, and pointed to from the table. A detailed description of the imaged area for all tested pairs should be included.

8) Both in the Abstract and in the Discussion section, the authors claim that they observe a small number of output channels. What is the exact evidence for this? The introduction and Results section could be made clearer by providing a definition of what authors consider as an input and an output to the central complex. Majority of the proposed input and output neurons seem to be related to the PB rather than the entire central complex. Thus, connectivity provided by the paper does not include inputs/outputs to the central complex from other neuropils. This should be made clearer.

---

## [Author Response]

1) The title overstates what is shown in the paper. The claim of a functional connectome raises the expectation that, if not all neurons, then at least all major neurons should be included in the analysis. Neither is the case. First, the main and best described input cells to the CX, the ring neurons connecting the bulbs to the ellipsoid body, are not included. Second, the pontine cells, numerically one of the most common neurons in the locust CX, are entirely missing from the study. Third, the input cells to the fan shaped body were largely excluded from the work as well. An introduction that better describes the anatomy of the CX and the cells examined in this study and a title that does not claim to build a functional connectome would improve this work.

We certainly agree with the reviewers that the central complex has many more cell types than we have tested for connections in this study. We mentioned this explicitly in the previous version of the manuscript but have now made this even clearer in Introduction and Discussion section. We describe CX anatomy in subsection “Cell types and hypothetical information flow in the central complex”, and have added details about the cell types considered/omitted (e.g., the pontine cells) in the first paragraph of the Results section.

We respectfully disagree with the reviewers about the title being overstated. In fact, we chose our title with the specific intention of avoiding overstatement. To us the present continuous tense of the first word in “Building a functional connectome of the *Drosophila* central complex” makes it clear that this is an ongoing effort rather than a complete connectome. If the reviewers feel strongly about this, we would be happy to change the title to “Towards a functional connectome of the *Drosophila* central complex”, which was another title we considered (but we prefer “Building” because the verb appropriately refers to the construction process that is described in the manuscript).

2) Many columnar neurons of the CX exist in more than one individual cell per column. Most extremely this was recently shown in bees, where 18 CPU4 cells (= P-FN cells) were shown to coexist in each column (Stone et al., 2017). This gives rise to the possibility that these cells could form circuits with each other rather than only with other types of neurons. As this appears to be a theme repeated for many cell types of the CX, this adds an entire layer of complexity to the CX-connectome that is left unexplored in the current paper. As any given driver line might only label a subset of anatomically identical cells (as shown for the P-EN cells), it would be key for interpreting the data, if the authors could provide information as to how many neurons per column are included in their drivers (to get an idea of how many cells were missed, once the total numbers for each cell type become available via other means).

Table 2 now provides this information based on available data. We also explicitly mention this issue in Discussion section.

3) The stimulation protocol only allows activation of all cells in a given population (rather than individual cells). As one hallmark of CX neurons is their spatially distinct activity across each population, this is a highly unnatural activity pattern, that can be expected to mask patterns of postsynaptic recruitment. The authors do mention this in the discussion and address the issue of recruiting global inhibitory circuits. The control for this with picrotoxin is an essential step in the right direction, but the possibility of non-GABAergic inhibition is looming in the background. This is especially important as the emphasized bottleneck of the CX circuit (the Delta7 cells) have been described as peptidergic (Allatostatin expressing) in other insects (TB1 cells are the homologous cells in all other insects). Inhibition mediated by this peptide has recently been shown to play an important role in mediating sleep in the CX of Drosophila (10.1016/j.neuron.2017.12.016). The conclusions drawn from the lack of direct columnar neuron connections within the PB are therefore to be taken with caution and needs more extensive discussion.

There is no report of AstA staining in the PB of *Drosophila* (https://doi.org/10.1016/j.neuron.2017.12.016, https://doi.org/10.1371/journal.pgen.1006346 and https://doi.org/10.1002/cne.22520). Of course, this does not preclude the involvement of other peptides or of metabotropic transmission. We mention this in subsection “Limitations of the method”.

4) Even though the focus of the work lies on Drosophila, much of its importance results from the broad impact of the paper for other insects. The discussion should be modified to place the work in context with studies from other insects. There is a lot known about the neurons constituting the described input and output pathways of the CX in locusts, bees, beetles and butterflies. Also, the homologous neuron of Delta7 (TB1) has been known to play a key role in the compass circuit of locusts for a decade and has been described in much detail (e.g. 10.1002/cne.23736, 10.1126/science.1135531). In fact, the described connectivity (indirect link between E-PG and columnar output cells via Delta7) in the current paper was also a prediction of Stone et al., (2017) in bees. There, this neuron was instrumental in constructing a ring attractor model needed to generate a compass signal that was required as direction signal for a path integration circuit. While some differences exist between the detailed TB1/Delta7 morphologies, the overall pattern that defines the cell is highly conserved across all species. Similarly, the output cells shown as being the only CX-output in the current work, are equivalent to the CPU1 cells in other insects, which have been proposed to be the sole output of the compass circuit (or path integration circuit model) in those species as well.

Table 2 has been added to link types across species. The first paragraph of subsection “Central complex motifs” now mentions the TB1 neuron.

5) The CX shows state dependent neural activity, is involved in sleep/attention regulation, and is predicted to drive actions in a context dependent way. Thus, output from this region will almost surely depend on the behavioral and motivational state of the animal, which is not maintained in the ex-vivo preparation used in this paper. Synaptic transmission efficiency can be expected to be differentially modulated in different behavioral contexts when several inputs converge onto the output cells (as suggested e.g. by Stone et al., 2017). With this in mind, the finding that there is no strong input from any cell onto the only output cell of the system needs to be discussed.

We agree that the animal’s behavioral and internal state are likely to modulate the activity of CX neurons, as has been demonstrated in several studies. We have added a few sentences on this in the Limitations section of the Discussion section. However, a likelier reason for our not observing more connections from CX neurons to identified output neurons —in the lateral accessory lobe (LAL), for example— is our screen’s incomplete coverage, particularly of FB neurons. We have added an appropriately worded sentence to Discussion section.

Similarly, the baseline fluctuations that allowed the authors to identify inhibitory connections are likely a result of state-changes of the neural circuit. Yet it is unclear whether these occur in all neurons equally likely. The authors indeed hint that this might not be the case, so that it appears unclear as to how many inhibitory connections could have been overlooked.

Baseline distributions are shown in Figure 3—figure supplement 4. Most neurons have large enough baseline deviations for the detection of inhibition. We added a sentence to Discussion section detailing the two possible exceptions (P-EG and PF-LCre). We also mention state dependence in this context.

The ex-vivo nature of the experiments should be mentioned early in the paper and the potential implications of this rather unnatural state of the brain should be discussed more clearly.

Done.

6) Cell types and cell pairs tested: There is mention of 24 cell types, but it seems only 21 types were tested.

24 are tested, but only 21 appear in the diagram to keep it lean (the extra 3 are in the matrix, have few tested partners and no detected connections). This is now clarified in the figure legend.

It is also unclear which percentage of potential synaptic partners were tested- are there neurons which overlap in their arbors, but were not yet tested? In addition, is it possible to roughly estimate the percentage of central complex neurons covered by this screen? If yes, the authors should include this information in the paper.

We now give the proportion of types described in Wolff, (2015) at the beginning of the Results section. For other types this is more difficult to evaluate.

7) A detailed description of each tested pair with the genotype is necessary and could be included as a table. Information about the exact Gal4-LexA pairs that are tested, the full genotypes of each tested pair, the pre and post synaptic cell names attached, the neuropils each cell occupies, and the observed connectivity would provide a great "cheat sheet". In addition, image stacks for the tested pairs should be made available online, if not already available, and pointed to from the table. A detailed description of the imaged area for all tested pairs should be included.

We now give the proportion of types described in Wolff, (2015) at the beginning of the result section. For other types this is more difficult to evaluate.

8) Both in the Abstract and in the Discussion section, the authors claim that they observe a small number of output channels. What is the exact evidence for this? The introduction and Results section could be made clearer by providing a definition of what authors consider as an input and an output to the central complex. Majority of the proposed input and output neurons seem to be related to the PB rather than the entire central complex. Thus, connectivity provided by the paper does not include inputs/outputs to the central complex from other neuropils. This should be made clearer.

We have added a clarifying sentence on this to the Introduction.

Inputs described include the EB, PB and NO. We clarified further in the introduction and result section that the FB was not included.

It is true that the current dataset has many neurons related to the PB —a result of the availability of clean and well-characterized lines for neuron types innervating the structure. We now clarify what we mean by inputs and outputs in the Discussion section and also mention the PB bias of the current dataset.